# FOVAL: Calibration-Free and Subject-Invariant Fixation Depth Estimation Across Diverse Eye-Tracking Datasets

**Anonymous**

## Abstract

Accurate fixation depth estimation is essential for applications in extended reality (XR), robotics, and human-computer interaction. However, current methods depend heavily on user-specific calibration, limiting their scalability and usability. We introduce FOVAL, a robust calibration-free approach that combines spatiotemporal sequence modelling via Long Short-Term Memory (LSTM) networks with subject-invariant feature engineering and normalisation. Compared to Transformers, Temporal Convolutional Networks (TCNs), and CNNs, FOVAL achieves superior performance, particularly in scenarios with limited and noisy gaze data. Evaluations across three benchmark datasets using Leave-One-Out Cross-Validation (LOOCV) and cross-dataset validation show a mean absolute error (MAE) of 9.1 cm and strong generalisation without calibration. We further analyse inter-subject variability and domain shifts, providing insight into model robustness and adaptation. FOVAL's scalability and accuracy make it highly suitable for real-world deployment.

## 1  Introduction

Fixation depth estimation plays a central role in applications such as extended reality (XR), robotics, autofocal eyewear, and human-computer interaction. Accurate predictions of fixation depth allow systems to adapt visual content to user attention, support immersive interactions, and enhance assistive technologies for vision. Despite ongoing advances, most existing methods remain dependent on user-specific calibration, which limits scalability and generalizability. Classical geometric and disparity-based approaches are prone to sensor noise and quantisation errors [5, 21], while data-driven models often require personalised calibration or struggle to generalise across diverse user populations [28, 18]. Recent hybrid models, such as Mix-TCN [33], integrate deep learning with geometric priors like Vestibulo-Ocular Reflex (VOR) constraints, and have shown strong performance in structured settings. However, their reliance on calibration-based preprocessing undermines their practicality in uncontrolled, real-world environments. To overcome these challenges, we introduce FOVAL, a calibration-free model for fixation depth estimation. FOVAL utilises Long Short-Term Memory (LSTM) networks to capture spatiotemporal gaze dynamics, supported by subject-invariant preprocessing and carefully engineered feature representations optimised for depth inference.

The core contributions of FOVAL are as follows. First, we present a subject-invariant processing pipeline that combines global and subject-wise normalisation to mitigate physiological and behavioural variability between individuals. Second, we develop a robust feature set that includes vergence-based metrics and dynamic gaze descriptors tailored to depth prediction. Third, we conduct comprehensive ablations demonstrating that LSTM-based models outperform Transformer, GRU, CNN, and TCN architectures in conditions characterised by limited and noisy data. Finally, we validate FOVAL across three diverse datasets using Leave-One-Out Cross-Validation (LOOCV) and cross-dataset transfer, showing consistent generalisation and achieving a mean absolute error (MAE) of approximately 9 cm without user-specific calibration. These results establish FOVAL as a practical,

Preprint.

scalable solution for calibration-free gaze-based depth estimation, advancing the deployment of gaze-enabled systems in everyday environments.

**Foveal Attention LSTM**

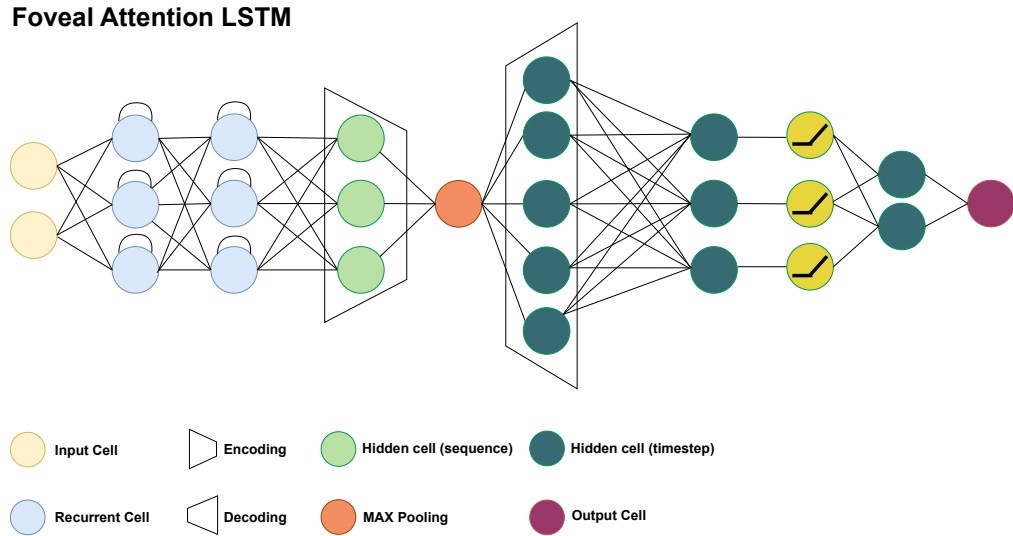

Figure 1: Schematic representation of the FOVAL model architecture. The architecture begins with an LSTM layer (hidden size=1435) to capture temporal dependencies, followed by batch normalisation for stabilisation. A max-pooling layer then identifies the most critical timestep, acting as an information bottleneck that distils the sequence's essence. This key timestep is processed through a dropout layer (probability p=0.245) to mitigate overfitting. Subsequent fully connected layers progressively expand and shrink the dimensionality (to 1763 and then to 440). This enables the model to explore and identify complex patterns within the critical timestep before the final prediction is made through an exponential linear unit (ELU).

## 2   Related Work

Calibration-free gaze estimation has gained momentum in recent years, aiming to reduce reliance on explicit, user-specific calibration routines. Auto-calibration techniques leverage environmental depth cues (often captured by RGB-D sensors) to dynamically align gaze estimations [13, 28]. Implicit calibration approaches, meanwhile, rely on anatomical regularities and gaze-centring assumptions. Self-supervised learning methods [31, 30, 10, 32] have shown promising results in eliminating the need for calibration in 2D gaze tasks. However, these strategies have yet to extend effectively into depth estimation, where noise, occlusion, and viewpoint variation pose additional challenges. FOVAL fills this gap by directly targeting depth estimation through robust temporal modelling and subject-invariant preprocessing.

Vergence geometry models draw from the biological principle that convergence of the eyes corresponds to focal depth. Techniques such as phase-based vergence control [24] and microsaccadic corrections [3] have demonstrated high fidelity in controlled settings, though their reliance on fine-grained resolution and stable viewing conditions constrains real-world applicability [21, 17]. FOVAL incorporates vergence-inspired features, such as eye alignment and directional disparity, while avoiding strict calibration dependencies.

Spatiotemporal neural networks have proven effective for sequence modelling, with models such as LSTMs, GRUs, and TCNs being widely adopted across gaze estimation tasks. Notably, Mix-TCN [33] combines bidirectional LSTMs with causal convolutions and achieves strong results in depth estimation. Yet, its dependence on VOR-based calibration and the unavailability of reproducible evaluation benchmarks limit comparative research. In contrast, FOVAL uses a fully calibration-free pipeline, validated on diverse datasets with publicly available code to ensure reproducibility and accessibility.

Recent efforts have also explored the application of Transformers in sequence-based gaze tasks [26, 29]. Although they exhibit strong representational power, their performance deteriorates with smaller, noisier datasets, which are common in gaze-based depth estimation. Our empirical findings confirm that LSTM-based models maintain higher accuracy and stability under such conditions.

Domain adaptation techniques are increasingly important in gaze estimation, especially given the sensitivity of models to shifts in hardware, task structure, and environmental context. While adversarial and clustering-based strategies [25, 4, 14] have improved cross-domain performance in 2D gaze tasks, they remain underexplored in the context of 3D depth inference. FOVAL explicitly integrates domain-aware normalisation and alignment techniques, improving generalisation across diverse recording setups.

Fixation depth estimation has traditionally been framed through geometric triangulation [5], stereo gaze, or monocular VOR-based models [16]. While effective in constrained scenarios, such approaches typically require static calibration or controlled task setups. More recent multitask frameworks [8] combine gaze regression with auxiliary supervision to improve performance, though they often retain calibration dependencies. FOVAL advances this line of work by enabling calibration-free, single-task depth inference without compromising accuracy.

Together, these strands of literature establish the landscape in which FOVAL operates. By uniting temporal learning, vergence-inspired features, and domain-robust preprocessing, FOVAL delivers a practical and generalisable solution to the problem of calibration-free fixation depth estimation.

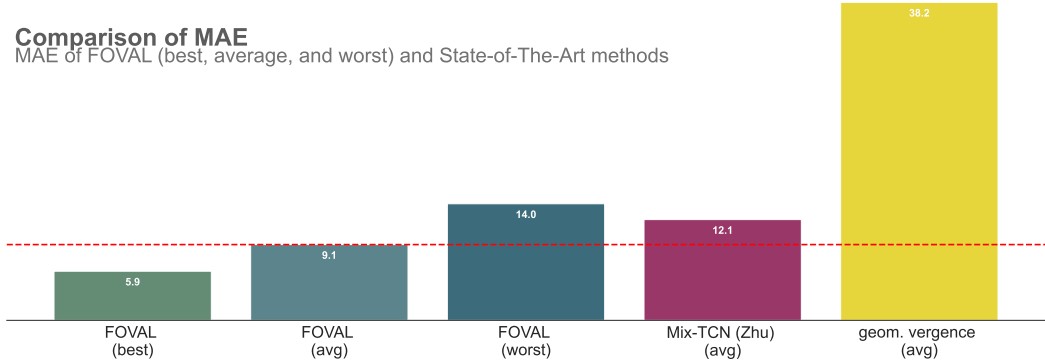

Figure 2: Direct comparison of FOVAL performance with the Mix-TCN model [33] and traditional vergence methods. FOVAL demonstrates a significantly lower MAE, averaging around 9.1 cm, with best-case performances reaching 5.9 cm.

## 3   Methodology

We formulate the fixation depth estimation task as a temporal regression problem. Given eye-tracking data at time $t$, represented as a feature vector $\mathbf{x}_t \in \mathbb{R}^d$, the objective is to estimate the corresponding fixation depth $y \in \mathbb{R}$ from a sequence $\mathbf{X} = \{\mathbf{x}_1, \mathbf{x}_2, \ldots, \mathbf{x}_T\}$ . The goal is to learn a function $f_\theta : \mathbb{R}^{T \times d} \to \mathbb{R}$, minimising the loss function:

$$\mathcal{L}(\theta) = \frac{1}{N} \sum_{i=1}^{N} \ell(\hat{y}_i, y_i^*),$$

where $\ell$ is the Smooth L1 Loss, chosen for its robustness to outliers and sensitivity to small errors.

To enable robust and generalisable learning, FOVAL employs a two-stage preprocessing pipeline as described in Appendix A. First, features undergo global robust scaling to mitigate outlier influence and normalise distributions across the full training set. Then, a subject-wise normalisation step is applied to reduce physiological variability, enhancing generalisation across users with differing eye anatomy or gaze behaviour.

The feature representation is tailored to capture the geometric and dynamic aspects of depth perception. Core features include eye vergence angles, eye direction vectors, and their angular differences, as well as derived metrics such as directional magnitudes, gaze acceleration, and ratios of gaze component velocities. These are further refined using logarithmic and Box-Cox transformations to ensure approximately Gaussian distributions, which support stable optimisation during training. To account for noise and artefacts in real-world gaze data, we incorporate a robust outlier removal mechanism based on the interquartile range (IQR) and a rolling mean window filter. These steps help suppress short-term spikes and preserve valid patterns in temporal sequences.

The overall architecture of FOVAL is illustrated in Figure 1 and consists of a single-layer Long Short-Term Memory (LSTM) network [6], selected for its proven ability to model long-range temporal dependencies in sparse and noisy data. The LSTM outputs are passed through batch normalisation [2] and a max-pooling operation [12] to extract the most informative timestep across the sequence. This key timestep is then processed by two fully connected layers with exponential linear unit (ELU) activation and dropout regularisation [22] to prevent overfitting. A linear output unit produces the final prediction.

We explored multiple architectures and found that LSTMs consistently outperform GRUs, CNNs, and Transformer variants in our setting, especially under conditions of limited training data and high noise. Hyperparameter tuning was conducted using Optuna [1] with a Tree-structured Parzen Estimator (TPE), optimising learning rate, hidden size, dropout, and weight decay.

Training was conducted using the AdamW [15] optimiser with a cosine annealing scheduler. Each model was trained for 500 epochs with early stopping and smooth L1 loss [20]. Input sequences were created using a sliding window over subject-wise time series, with a fixed length of ten timesteps, each comprising up to 38 features after transformation and filtering. This methodology yields a robust, scalable pipeline that enables calibration-free depth estimation with strong generalisation across users and domains.

## 4  Experimental Setup

To rigorously evaluate FOVAL's performance and generalisability, we conducted extensive experiments across three diverse gaze datasets. These datasets reflect varying acquisition conditions, hardware platforms, and user behaviours, allowing us to assess both within-dataset accuracy and cross-dataset robustness under realistic scenarios.

We selected the following benchmark datasets for evaluation: (1) The **Robust Vision (RV)** dataset [7] includes gaze recordings from 25 emmetropic participants (11 male, 14 female, $M = 31.7$ years, $SD = 11.7$) in a controlled virtual reality environment using the XTAL headset [27]. Participants fixated on a moving sphere at distances ranging from 0.35 to 3 meters, yielding 282,080 samples with precise gaze vectors and origins in headset coordinates. (2) The **Tufts Gaze Depth** dataset [23] was recorded in an indoor environment using the Pupil Labs Core eye tracker [19] and Intel RealSense D435 RGB-D camera [9], comprising over 75,000 samples. Participants fixated on 12 spatially distributed targets (1.9–6.4 m), producing 3D gaze data with ground-truth depth calibration. (3) The **Gaze-in-the-Wild (GIW)** dataset [11] captures free-form daily activities using mobile eye-tracking glasses paired with an IMU and ZED Mini stereo camera. It includes RGB+D imagery, infrared eye recordings, and head motion data, posing strong challenges to robustness due to environmental variability and natural head dynamics.

Together, these datasets span controlled to unconstrained conditions, short to long depth ranges, and stable to mobile recording setups—providing a comprehensive testbed for calibration-free depth estimation.

We adopt Mean Absolute Error (MAE) as our primary evaluation metric due to its interpretability and robustness against large deviations. To complement this, we include residual histograms, stratified threshold accuracy, and visual comparisons of predicted versus ground-truth depth. This layered evaluation offers both statistical rigour and practical insights.

All within-dataset evaluations are performed using subject-wise Leave-One-Out Cross-Validation (LOOCV), providing a strong test of generalisation to unseen individuals. For cross-dataset generalisation, we train on one dataset and validate on another, applying dataset-specific feature normalisation and aligning target depth ranges to mitigate distributional mismatch.

To contextualise performance, we compare FOVAL against alternative architectures including GRUs, 1D-CNNs, TCNs, and Transformer-based models. Each model is trained under identical preprocessing and LOOCV conditions. Cross-dataset experiments are restricted to FOVAL's LSTM variant, focusing on transferability rather than architectural variance.

This experimental framework enables systematic exploration of subject variability, domain shifts, and model robustness, which are core aspects for real-world deployment of calibration-free fixation depth models.

**Computing Resources**  All experiments were conducted on a workstation equipped with an NVIDIA RTX 5070 GPU (12 GB VRAM), AMD Ryzen 7 7700 CPU, and 32 GB RAM. Each LOOCV experiment (25 folds) took approximately 4-5 hours to complete, depending on the dataset. Total compute time across all experiments, including cross-dataset evaluations and ablations, was approximately 500 GPU-hours.

## 5    Results

We report results across within-dataset, cross-dataset, and architectural comparisons, followed by a detailed analysis of residual errors and inter-subject variability.

### 5.1    Within-Dataset Evaluation (LOOCV)

FOVAL achieves strong accuracy across all datasets with subject-wise Leave-One-Out Cross-Validation (LOOCV). Specifically, the model attains an MAE of 9.1 cm on the Robust Vision (RV) dataset, 9.7 cm on the Tufts dataset, and 9.2 cm on the Gaze-in-the-Wild (GIW) dataset (see Table 5.1). These consistent performances underscore FOVAL's ability to generalise across varied gaze conditions and task structures without requiring individual calibration. A direct comparison to the state-of-the-art Mix-TCN model and traditional vergence-based methods highlights FOVAL's performance advantage across all datasets (see Figure 2).

| Dataset | MAE (cm) | Notes |
|---|---|---|
| Robust Vision (RV) | 9.1 | Controlled VR environment |
| Tufts | 9.7 | Indoor 3D real-world scene |
| Gaze-in-the-Wild (GIW) | 9.2 | Naturalistic mobile conditions |
| Combined (All Datasets) | 18.0 | Strong domain shift |

Table 1: LOOCV Mean Absolute Error (MAE) of FOVAL across datasets. Lower is better.

### 5.2    Cross-Dataset Generalisation

When evaluated on unseen datasets, FOVAL maintains competitive performance. Training on RV and evaluating on Tufts and GIW yields MAEs of 11.81 cm and 11.96 cm, respectively. When trained on GIW and tested on Tufts, FOVAL reaches an MAE of 12.67 cm. These results confirm the method's robustness under domain shifts caused by different hardware, environmental conditions, and gaze dynamics. Notably, aligning depth ranges and applying dataset-specific normalisation significantly improve transferability compared to naive transfer setups (Table 5.2).

| Train Set | Test Set | Method | MAE (cm) |
|---|---|---|---|
| RV | Tufts | LOOCV | 11.81 |
| RV | GIW | LOOCV | 11.96 |
| Tufts | GIW | LOOCV | 19.02 |
| Tufts | RV | LOOCV | 16.52 |
| GIW | Tufts | LOOCV | 12.67 |

Table 2: Cross-dataset evaluation of FOVAL after depth range alignment and dataset-wise normalisation.

## 5.3 Prediction Confidence and Residuals

Post-hoc analysis reveals that 67.5% of predictions fall within 10 cm of the ground-truth depth, and 88.4% within 20 cm (Figure 4). The residual distribution is slightly skewed towards overestimation, with a mean residual error of -2.05 cm (Figure 3). Error magnitudes increase at extreme depth ranges, particularly below 0.5 m and beyond 3.5 m, reflecting the nonlinear nature of vergence behaviour at depth extremes. The optimal prediction range lies between 0.4 and 2.5 meters (Figure 5), where vergence is approximately linear. Errors increase outside this window.

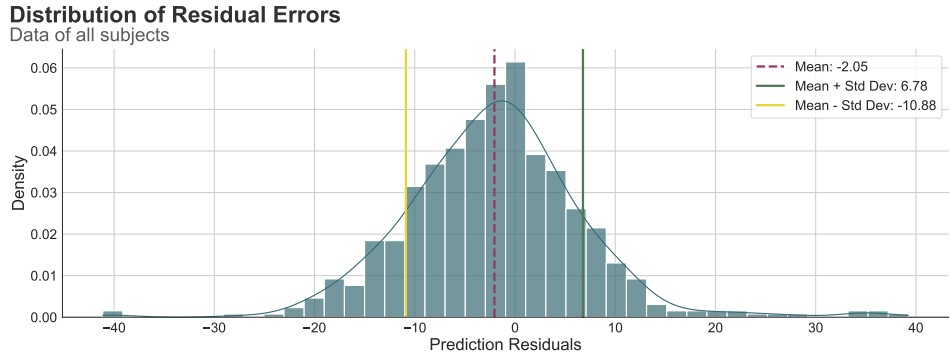

Figure 3: Histogram of residual errors for fixation depth predictions. The distribution shows a slight overestimation bias (mean: $-2.05$ cm).

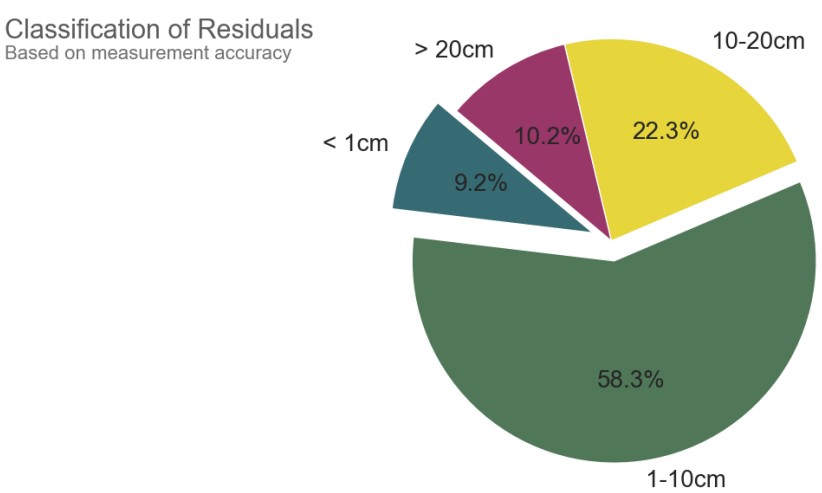

Figure 4: Distribution of prediction errors across four error size ranges. Notably, 67.5% of predictions are below 10 cm.

## 5.4 Inter-Subject Variability

Performance varies across participants. Best-case performance reaches 5.9 cm MAE, while the worst-case subject attains 14.61 cm. (Figure 6). This variability is likely influenced by differences in gaze behaviour and recording quality. Clustering analysis based on gaze dynamics suggests that subjects with stable vergence profiles consistently achieve lower errors, opening future opportunities for personalised adaptation (Figure 7).

## 5.5 Architecture Comparison and Ablation Summary

To validate the robustness of FOVAL's design, we benchmarked its LSTM backbone against alternative architectures including GRU, 1D-CNN, TCN, and Transformer-based models. Across all comparisons,

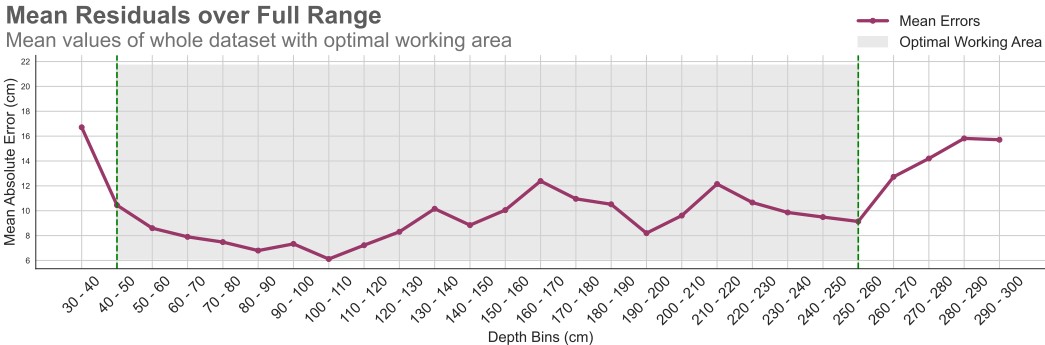

Figure 5: Average error across 10 cm depth bins. FOVAL performs best between 50 and 250 cm.

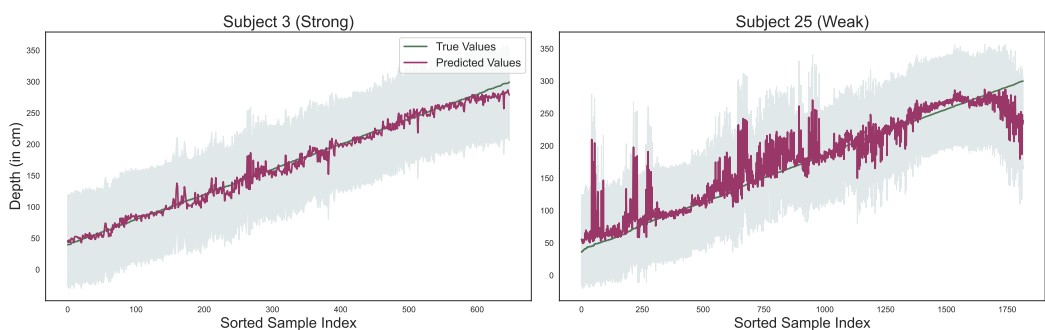

Figure 6: Prediction quality for two representative subjects. Left: Subject 3 (MAE = 5.9 cm); Right: Subject 25 (MAE = 14.0 cm).

the LSTM consistently achieved the lowest mean absolute error (MAE) under real-world, noisy conditions. Ablation experiments further showed that removing batch normalisation or replacing the LSTM with a GRU substantially degraded performance. These results reinforce our architectural choices. Full performance metrics and architectural ablation details are provided in Appendix C. We benchmarked FOVAL's LSTM backbone against GRU, TCN, 1D-CNN, and Transformer models. The LSTM consistently outperformed all alternatives in terms of robustness and accuracy. Detailed results and ablations are provided in Appendix C.

To assess the robustness of our findings, we performed Wilcoxon signed-rank tests across subjects comparing LSTM with alternative architectures. Results confirm that FOVAL's LSTM significantly outperforms Transformer ($p < 1e-9$) and TCN ($p < 0.001$), with trends over GRU and no significant difference to 1D-CNN (see Appendix D).

> **Application Example: Autofocal Glasses and Real-World Gaze Interfaces**
>
> FOVAL enables real-time, calibration-free fixation depth estimation, making it highly suitable for practical deployment in wearable devices such as **autofocal glasses**, where dynamic focal plane adaptation is essential for visual comfort. Its subject-invariant design and robust performance under noisy, mobile, or consumer-grade conditions open new possibilities for **gaze-driven interaction in AR/VR**, vision correction systems, and attention-aware robotics.

## 6 Discussion

FOVAL presents a significant step forward in the development of calibration-free systems for fixation depth estimation. Its strong performance across three diverse datasets demonstrates that subject-

specific calibration, long considered a necessity in gaze-based depth estimation, can be effectively bypassed with the right combination of feature engineering, subject-invariant preprocessing, and robust sequence modelling.

Our within-dataset evaluations highlight the model's ability to deliver sub-10 cm accuracy consistently, making it suitable for practical applications in XR, vision enhancement, and human-computer interaction. The performance of 9.1–9.7 cm MAE across datasets (Table 5.1) is particularly noteworthy given the heterogeneity in hardware setups and recording protocols.

When challenged with domain shifts in cross-dataset settings, FOVAL continues to perform robustly, achieving transfer MAEs of 11.8–12.6 cm. These results affirm that the model is not overfitting to a specific domain or user group, but instead learns generalisable patterns in gaze dynamics. Our finding further supports this, that proper depth range alignment and dataset-wise normalisation significantly improve generalisation.

From an architectural perspective, LSTM emerges as the most reliable sequence model under real-world conditions. Despite the popularity of Transformers, their poor performance (84.65 cm MAE) underscores their unsuitability for low-volume, noisy datasets. The LSTM architecture strikes a strong balance between expressivity and robustness, with ablations confirming that both temporal modelling and batch normalisation are essential to achieving stable convergence and low error rates.

In addition to aggregate accuracy, the residual analyses provide insights into model bias and behaviour. A mild overestimation tendency was observed (Figure 3), especially at extreme depth ranges. These systematic deviations could be addressed in future iterations using adaptive uncertainty estimation or non-linear output calibration techniques. Moreover, the fact that nearly 70% of predictions fall within a 10 cm error window (Figure 4) speaks to the model's practical utility in real-world settings.

Inter-subject variability remains a critical challenge. The MAE ranged from 5.9 cm to over 14 cm across subjects, suggesting that personalised fine-tuning or clustering-based adaptation might further improve performance. Our initial analysis of gaze dynamics (Figure 7) supports the feasibility of such approaches, revealing clusters of users with shared gaze characteristics and distinct error profiles.

Overall, FOVAL offers a generalisable and accurate solution for fixation depth estimation that is resilient across users, datasets, and hardware platforms. The modularity of its design and open-source codebase support its adoption in varied real-world systems, from adaptive XR displays to mobile health diagnostics, paving the way for more inclusive and scalable gaze-based technologies.

## 6.1   Limitations

While FOVAL provides a robust baseline, this study is constrained by limitations in dataset diversity. None of the datasets used include outdoor environments, participants with visual impairments, or a broad demographic variety. Expanding coverage to dynamic, uncontrolled environments and medically diverse populations is a necessary step toward clinical and consumer applications.

# 7   Future Directions

Building on FOVAL's strengths, several key directions merit exploration. First, we will expand our dataset to include outdoor recordings and visually impaired populations to test generalizability in real-world and clinical contexts. Second, integrating adaptive attention mechanisms and uncertainty-based loss functions may help the model learn personalised representations while maintaining robustness.

Additionally, transfer learning techniques, such as fine-tuning on small in-domain batches, could allow deployment in new domains without retraining from scratch. Finally, we strongly advocate for the creation of open benchmarks and annotated datasets to foster reproducibility, transparency, and shared progress across the field.

Beyond XR and assistive eyewear, fixation depth estimation has growing relevance in domains such as driver attention modelling in automotive interfaces, where continuous gaze depth can help detect fatigue or distraction. Similarly, clinical use cases such as assessing strabismus, depth perception anomalies, or progressive oculomotor disorders would benefit from calibration-free approaches deployable on consumer-grade eye trackers

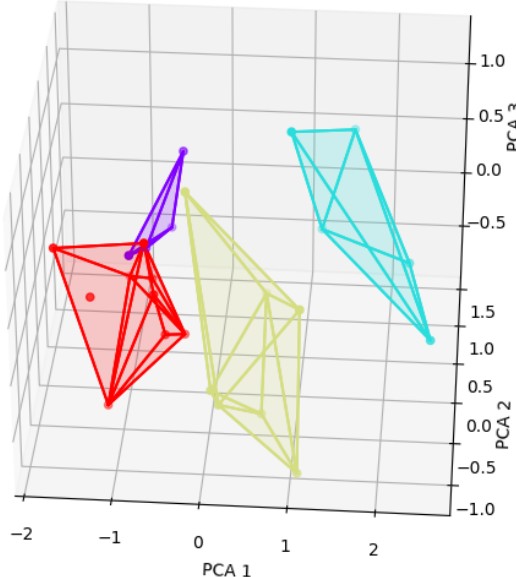

Figure 7: Subject clustering based on gaze dynamics and physiological metrics using PCA. Cluster A (lower MAE) shows stable vergence behavior, while Cluster B (higher MAE) reveals greater variability in gaze and IPD. These patterns motivate future work on adaptive or personalised modelling.

## Impact Statement

FOVAL represents a transformative step toward accessible, scalable, and calibration-free gaze depth estimation. By eliminating the need for subject-specific calibration and functioning reliably on consumer-grade hardware, FOVAL democratizes access to gaze-enabled technologies in extended reality (XR), vision health, and human-computer interaction.

However, the increasing use of gaze data also raises critical ethical concerns. Gaze reveals attention, intention, and preference. Highly sensitive behavioural cues that can be exploited. To ensure responsible adoption, we call for the implementation of privacy-preserving algorithms and transparent usage guidelines. Gaze features should be anonymised or obfuscated where possible, and consent mechanisms must be embedded into data collection systems.

We encourage collaborations between technologists, ethicists, and policymakers to co-create standards that guide the ethical deployment of gaze-based technologies. If done correctly, gaze tracking can improve accessibility, enable adaptive systems, and support health diagnostics without compromising individual rights.

## Code and Data Availability

To foster transparency and reproducibility, we release our code, training pipeline, and preprocessed datasets used in this study at: `https://anonymous.4open.science/r/foval-4E13/` . Instructions on how to reproduce all experiments and evaluation scripts are provided in the repository. The used datasets (Robust Vision, Tufts, GIW) are publicly available, and we provide detailed configuration files to replicate each experiment.

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

## Appendix Overview

This appendix provides supporting details to ensure reproducibility of our experiments with FOVAL. It includes a comprehensive description of preprocessing steps, engineered features, and the training setup. A summary of key hyperparameters used for the final model is also provided.

## A    Preprocessing and Feature Engineering

We employed a multistage preprocessing pipeline to improve model robustness and generalisation. This section describes how we cleaned, transformed, and normalised the data, along with the feature computations used as input for the model.

### A.1    Data Cleaning

To identify and remove anomalies in the dataset, we employed a rolling mean window approach with a window size of 5 and a threshold of 10. This technique captures context-specific, short-term anomalies that might arise due to noise in the input signal. The process for detecting anomalies is formalised as follows:

$$W_i = \{GT_{depth_{\text{start}}}, \ldots, GT_{depth_{\text{end}}}\} \tag{1}$$

where

$$\text{start} = \max\left(i - \frac{\text{window}_{size}}{2}, 0\right) \tag{2}$$

$$\text{end} = \min\left(i + \frac{\text{window}_{size}}{2} + 1, N\right) \tag{3}$$

To identify outliers within our feature space, we employed the Interquartile Range (IQR) method, where $Q1$ and $Q3$ represent the first and third quartiles:

$$IQR = Q3 - Q1 \tag{4}$$

Outliers are defined as:

$$\text{Outlier if:} \quad \text{df}_{\text{column}} > Q3 + 1.5 \times IQR \quad \text{or} \quad \text{df}_{\text{column}} < Q1 - 1.5 \times IQR \tag{5}$$

This process ensures that outliers, which can skew the data and reduce model accuracy, are effectively removed.

### A.2    Data Balancing

To balance the data, we divided the target variable, $GT_{depth}$, into 10 cm bins. This ensures that the distribution of depth values across the dataset is balanced, which prevents bias in model predictions.

The binning process is formalized as:

$$GT_{depth_{bin}} = \text{Bin}(GT_{depth}, 60) \tag{6}$$

Next, to ensure that each bin had a similar number of samples, we performed resampling:

$$\bar{C} = \frac{1}{N_{bins}} \sum_{i=1}^{N_{bins}} C_i \tag{7}$$

We applied oversampling for bins with fewer samples and undersampling for bins with excess samples to achieve balance. The resampling process is described as:

$$D_i' = \begin{cases} \text{Resample}(D_i, \text{replace} = \text{True}, n = \lceil \bar{C} \rceil), & \text{if } |D_i| < \bar{C} \\ \text{Resample}(D_i, \text{replace} = \text{False}, n = \lceil \bar{C} \rceil), & \text{if } |D_i| > \bar{C} \\ D_i, & \text{otherwise} \end{cases} \quad (8)$$

### A.3 Data Splitting

We used K-Fold Cross-Validation (KFold) with $n = 25$ (the number of subjects) to ensure robust evaluation across different subjects. To avoid data leakage, the dataset was split on a per-subject basis, meaning no data from the same subject appeared in both the training and testing sets.

$$KFold(\text{splits} = \text{n}_{\text{splits}}, \text{shuffle} = \text{True}, \text{random state} = 42) \quad (9)$$

This ensures that the model's performance is evaluated on entirely unseen data, mimicking real-world applications where generalizability is crucial.

### A.4 Feature Engineering

This section describes our specific computations for each feature in the feature engineering process.

**Eye Vergence Angle (EVA)**   The vergence angle ($\theta$) between the eyes is calculated as:

$$\theta = \cos^{-1} \left( \frac{\mathbf{v}_L \cdot \mathbf{v}_R}{|\mathbf{v}_L||\mathbf{v}_R|} \right)$$

Where $\mathbf{v}_L$ and $\mathbf{v}_R$ represent the gaze direction vectors of the left and right eyes, respectively. This provides insight into the focal depth and 3D perception.

**Eye Direction Vectors (EDV)**   We compute the Euclidean distance between the focus points of both eyes to determine the convergence or divergence of the gaze. These vectors provide additional data about the accuracy of depth perception.

**Interpupillary Distance (IPD)**   The IPD is calculated as the Euclidean distance between the gaze origins of the right and left eyes:

$$IPD = \sqrt{(\Delta X)^2 + (\Delta Y)^2 + (\Delta Z)^2} \quad (10)$$

where $\Delta X$, $\Delta Y$, and $\Delta Z$ are the differences in the $X$, $Y$, and $Z$ coordinates of the right and left eye gaze origins, respectively.

**Vergence Angle**   The vergence angle (VA) between the gaze directions of the right and left eyes is calculated using the dot product, normalized by the magnitudes of the gaze direction vectors:

$$\text{VA} = \arccos \left( \frac{\vec{R} \cdot \vec{L}}{\|\vec{R}\|\|\vec{L}\|} \right) \quad (11)$$

where $\vec{R}$ and $\vec{L}$ are the gaze direction vectors for the right and left eyes, respectively.

**Normalized Vergence Angle**   Normalizing the vergence angle between -1 and 1 provides a standardized scale for this feature, making the model's learning process less sensitive to the absolute scale of the original angles. It's calculated as:

$$\text{VA}_{\text{normalized}} = 2 \left( \frac{\text{VA} - \min(\text{VA})}{\max(\text{VA}) - \min(\text{VA})} \right) - 1 \quad (12)$$

**Vergence Depth Calculation**    Vergence Depth (VD) is computed based on the vergence angle and the IPD, assuming a simple geometric model of eye vergence:

$$\text{VD} = \frac{IPD}{2\tan\left(\frac{\text{VA}}{2}\right)} \tag{13}$$

This depth is then converted from meters to centimeters. Please note that specific calculations such as velocity, acceleration, and angular differences require sequential data points and depend on the temporal resolution of the data collected.

**Normalized Depth**    The normalized depth for each observation is calculated as:

$$\text{VD}_{\text{normalized}} = \frac{\text{VD} - \min(\text{VD})}{\max(\text{VD}) - \min(\text{VD})} \tag{14}$$

**Directional Magnitude**    The magnitude of gaze direction vectors for the right and left eyes are computed using the Euclidean norm:

$$\vec{D}_R = \|\vec{R}\| \tag{15}$$

$$\vec{D}_L = \|\vec{L}\| \tag{16}$$

where $\vec{R}$ and $\vec{L}$ are the gaze direction vectors for the right and left eyes, respectively.

**Gaze Direction Cosine Angles**    The cosine of vergence angles, derived from the eye vergence angle data, represents the orientation difference between the gaze directions of the two eyes. This feature is essential because it captures the degree of alignment or disparity between the eyes' focus, which can indicate depth perception and the point of focus in 3D space. The cosine of the vergence angle is calculated as:

$$\text{Cosine Angles} = \cos(\text{VA}) \tag{17}$$

**Gaze Point Distance**    This feature calculates the Euclidean distance between the directions of the right and left eyes using their X and Y components. This distance can indicate how convergent or divergent the gaze is, which, like the vergence angle, relates to depth perception and focus. The Euclidean distance between the gaze points of the right and left eyes is computed as:

$$\text{POR}_{\text{Distance}} = \sqrt{(\vec{R}_X - \vec{R}_X)^2 + (\vec{R}_Y - \vec{R}_Y)^2} \tag{18}$$

where $\vec{R}_X$ represents the gaze direction of the right eye in the x direction.

**Angular Difference between Gaze Directions**    The angular difference between the gaze directions is calculated using the dot product and magnitude of gaze direction vectors:

$$\vec{D}_{\text{Angle}} = \arccos\left(\frac{\vec{D}_R \cdot \vec{D}_L}{\|\vec{D}_R\|\|\vec{D}_L\|}\right) \tag{19}$$

**Velocity and Acceleration**    Velocity and acceleration are derived from the gaze direction X-component (similar computations apply for Y and Z components) and the left eye:

$$\text{Velocity}_X = \Delta\vec{R}_X \tag{20}$$

$$\text{Acceleration}_X = \Delta\text{Velocity}_X \tag{21}$$

**Gaze Direction Ratios** $(\vec{R}_{X,Y,Z})$

$$R_X \& = \frac{\vec{R}}{L_X} \tag{22}$$

$$R_Y \& = \frac{\vec{R}}{L_Y} \tag{23}$$

$$R_Z \& = \frac{\vec{R}}{L_Z} \tag{24}$$

**Ratio of Delta Gaze** $(R_{\Delta Gaze_{XY}})$

$$R_{\Delta Gaze_{XY}} = \frac{R_{\Delta Gaze_X}}{R_{\Delta Gaze_Y}} \tag{25}$$

**Gaze Direction Angle**  The angle between the gaze direction vectors of the right and left eyes:

$$\vec{D}_{\text{Angle}} = \arccos\left(\frac{\vec{R} \cdot \vec{L}}{\|\vec{R}\|\|\vec{L}\|}\right) \tag{26}$$

**Relative Change in Vergence Angle**  The change in vergence angle over time (or between sequential observations):

$$\text{Relative Change VA} = V_t - V_{t-1} \tag{27}$$

**Ratio of Directional Magnitudes**  The ratio of the directional magnitudes (norms of gaze direction vectors) for the right and left eyes:

$$\text{Ratio}_{\vec{D}} = \frac{\|\vec{D}_R\|}{\|\vec{D}_L\|} \tag{28}$$

**Gaze Point Depth Difference**  The difference in depth (Z-coordinate) between the right and left gaze points:

$$\Delta \text{Gaze Point}_{Depth} = R_Z - L_Z \tag{29}$$

**Ratios of World Gaze Direction Components**  The ratios of corresponding components of the world gaze direction vectors for the right and left eyes:

$$\text{Ratio } \vec{D}_{World,X} = \frac{R_X}{L_X} \tag{30}$$

$$\text{Ratio } \vec{D}_{World,Y} = \frac{R_Y}{L_Y} \tag{31}$$

$$\text{Ratio } \vec{D}_{World,Z} = \frac{R_Z}{L_Z} \tag{32}$$

**Velocity and Acceleration**  The rate of change (velocity) and the rate of change of the rate of change (acceleration) in a given component (e.g., X) of the gaze direction:

$$\text{Velocity}_X = X_t - X_{t-1} \tag{33}$$
$$\text{Acceleration}_X = \text{Velocity}_{X_t} - \text{Velocity}_{X_{t-1}} \tag{34}$$

**Angular Difference X**  The difference in the X-component (horizontal) of the angular gaze direction between the right and left eyes:

$$\text{Angular Difference}_X = \theta_{RX} - \theta_{LX} \tag{35}$$

## A.5 Normalization

**Global Normalization**  We applied a Robust Scaler to normalize each feature across all subjects:

$$X' = \frac{X - \text{Median}(X)}{\text{IQR}(X)}$$

This ensures that outliers have minimal influence on the overall data distribution.

**Subject-wise Normalization** After global normalization, we applied a subject-wise normalization to account for individual variations:

$$X'_{s,\text{normalized}} = \frac{X'_s - \text{Median}(X'_s)}{\text{IQR}(X'_s)}$$

This two-step normalization process reduces bias and improves model generalization.

### A.6 Feature Transformation

To improve the normality of the feature distribution, we applied various transformations, including logarithmic and Box-Cox transformations:

**Logarithmic Transformation** For positively skewed features, we applied the log transformation:

$$X' = \log(X + 1)$$

**Box-Cox Transformation** For other features, we used the Box-Cox transformation:

$$X' = \frac{X^\lambda - 1}{\lambda}, \quad \text{where } \lambda \text{ is optimized for each feature.}$$

### A.7 Sequence Creation for Time-Series Analysis

For time-series analysis, we structured the dataset into overlapping sequences of 10 consecutive time steps, with each time step representing 12 input features.

For each subject $s$, the sequence at time $n$ is defined as:

$$S_n = \{X_n, X_{n+1}, \ldots, X_{n+L-1}\}$$
$$y_n = y_{n+L}$$

This sequence-based structure allows the model to capture temporal dependencies between eye movements.

## B Training Setup and Hyperparameters

The final FOVAL model was trained using the configuration in Table 3. We used Smooth L1 Loss and the AdamW optimiser with a cosine annealing schedule.

Table 3: Final training configuration for FOVAL.

| Parameter | Value |
|---|---|
| LSTM Hidden Units | 1435 |
| Fully Connected Layer (FC1) | 1763 |
| Dropout Rate | 0.245 |
| Batch Normalisation | Yes |
| Learning Rate | 0.0327 |
| Weight Decay | 0.0907 |
| Optimiser | AdamW |
| Scheduler | Cosine Annealing ($T_{\max} = 100$) |
| Loss Function | Smooth L1 Loss |
| Epochs | 500 |
| Input Sequence Length | 10 steps |

# C Model Architecture Comparisons

To evaluate the effect of model architecture on performance, we compared FOVAL's LSTM-based implementation against alternative temporal sequence models, including Gated Recurrent Units (GRU), 1D Convolutional Networks (1D-CNN), Temporal Convolutional Networks (TCN), and Transformer-based models. All models were trained using the same preprocessing pipeline and evaluated using LOOCV on the RV dataset.

Table 4: LOOCV performance comparison of different sequence modelling architectures on the Robust Vision dataset.

| Architecture | Avg. MAE (cm) | Best Fold MAE (cm) |
|---|---|---|
| LSTM (FOVAL) | 9.1 | 5.9 |
| GRU | 18.58 | 9.20 |
| 1D-CNN | 16.34 | 8.82 |
| TCN | 14.75 | 7.58 |
| Transformer (Multi-Head) | 84.65 | 75.53 |

The LSTM consistently outperforms the other architectures under constrained data and noisy conditions. GRU and TCN models show moderate degradation, while CNNs and Transformer-based models perform substantially worse—likely due to higher data requirements and sensitivity to noise.

Additional ablation results show that removing batch normalisation increases MAE to 21.43 cm, highlighting its stabilising effect. These results validate the architectural choices made in FOVAL.

# D Statistical Significance of Architecture Comparison

We performed Wilcoxon signed-rank tests to compare FOVAL's LSTM architecture against alternative sequence models across all subjects in the Robust Vision dataset. Table 5 reports the signed-rank statistics and p-values.

| Comparison | Wilcoxon Statistic | p-value |
|---|---|---|
| LSTM vs Transformer | 0.0 | $2.33 \times 10^{-10}$ |
| LSTM vs TCN | 101.0 | $8.81 \times 10^{-4}$ |
| LSTM vs GRU | 179.0 | $7.08 \times 10^{-2}$ |
| LSTM vs CNN | 212.0 | $2.28 \times 10^{-1}$ |

Table 5: Wilcoxon signed-rank test comparing LSTM against other models. Results show significant differences for Transformer and TCN.

