# OpenReview forum: "Subject-Invariant Normalization: A Simple Principle for Robust Sequence Modeling"
_ICLR.cc/2026/Conference — ICLR 2026 Conference Withdrawn Submission_

### Official Review · Reviewer_zhUf · 2025-10-22

**Soundness:** 2
**Presentation:** 2
**Contribution:** 3
**Rating:** 4
**Confidence:** 3

**Summary:**

This paper investigates the task of fixation depth estimation. To address limitations in scalability and usability caused by user-specific calibration, it introduces a robust, calibration-free approach that models spatiotemporal sequences using a Long Short-Term Memory (LSTM) network with subject-invariant feature engineering. The authors conducted experiments on benchmark datasets and cross-dataset evaluations to validate the effectiveness of the proposed method, which demonstrates strong generalization without calibration.

**Strengths:**

1. The authors conducted extensive experiments on public datasets in both within-dataset and cross-dataset settings to evaluate the proposed method.

2. The source code and pre-trained weights are provided; this is a great step toward evaluating the proposed method.

**Weaknesses:**

1. The proposed method seems somewhat trivial. It is built on a Long Short-Term Memory (LSTM) network and lacks modules specifically designed for the task of fixation depth estimation.

2. The paper reports only quantitative experimental results and lacks a qualitative analysis of the model’s predictions.

3. The subject-invariant feature engineering and normalization are hard to understand. How can this approach prevent the need for user-specific calibration?

**Questions:**

Basically, I have no questions about this paper. My major concerns about this paper have listed in the Weaknesses. Could the authors explain how the proposed subject-invariant feature engineering and normalization prevent the need for user-specific calibration?

---

### Official Review · Reviewer_dGAg · 2025-10-29

**Soundness:** 2
**Presentation:** 1
**Contribution:** 2
**Rating:** 2
**Confidence:** 4

**Summary:**

This paper introduces FOVAL, a calibration-free, subject-invariant approach for estimating fixation depth from eye tracking data using an LSTM-based spatiotemporal sequence model. The method focuses on feature engineering and normalization to achieve strong generalization across subjects and datasets (RV, GazeCapture), reporting a mean absolute error (MAE) of 9.1 cm on the Robust Vision (RV) dataset. The authors contrast their LSTM approach with other sequence models (TCN, Transformer).

**Strengths:**

Eye tracking cross-platform, cross-condition, and cross-subect without calibration is a challenging problem and the authors bring ML to bear on addressing this problem.
Calibration-free and subject-invariant fixation depth estimation is a highly relevant and commercially important topic, particularly for Extended Reality (XR) applications.
The attempt to validate the model using both Leave-One-Out Cross-Validation (LOOCV) and cross-dataset testing is appreciated and is a necessary step to support the claim of subject-invariance.

**Weaknesses:**

The paper suffers from significant deficiencies in its empirical rigor, depth of analysis, and presentation.
The entire empirical argument for the proposed FOVAL architecture rests on a single table of performance metrics that appears in the appendix (Table 4), comparing mean MAE values against baselines. This is insufficient evidence and analysis (I am also unclear if the authors are claiming SOTA performance?)
The figures are simplistic. More complex, information-rich visualizations would help the reader interpret the model's behavior, not just summary plots. Key figures showing the model's errors (e.g., error heatmaps, error distributions, or detailed qualitative results) are necessary but are notably absent or relegated to a basic format.
The core claim of Subject-Invariant performance is not rigorously demonstrated. The reported 9.1 cm MAE lacks context and a deep statistical breakdown. It should go beyond reporting the mean and instead provide a full analysis of the distribution and source of errors. This includes a detailed breakdown of the MAE to confirm the claim of subject-invariance, rather than just reporting the average and the "Best Fold" MAE.
The paper's attempt at statistical comparison (Wilcoxon signed-rank tests in Appendix D) is the absolute bare minimum and does not constitute interesting or insightful analysis.
The technical approach of using an LSTM network with feature engineering is incremental. The authors fail to articulate a compelling novel learning principle or architectural breakthrough beyond normalizing per-subject. The contribution seems to be a combination of existing sequence modeling and specific feature engineering/normalization, which makes the work feel more like a competent technical report on an application rather than a strong algorithmic contribution to deep learning

**Questions:**

Figure 1 This is a very different design than the standard LSTM diagram. Please explain how your model differs from a stock LSTM. Including a diagram of how various types of the gaze data included are fed into the model through time would bring more clarity overall
Table 4 is the main result interesting to the ICLR community. This should be included in the main text
“We explored multiple architectures and found that LSTMs consistently outperform GRUs, CNNs, and Transformer variants in our setting, especially under conditions of limited training data and high noise” Refer to the table with these results
The list of features chosen should be included as a table in the main paper. Include ablation studies to evaluate which of these are necessary and sufficient to achieve SOTA results.
To support the "Subject-Invariant" claim, include a figure (perhaps box plot or histogram) of the MAE across all subjects/LOOCV folds for FOVAL and the best baseline. Analyzing the inter-subject variance in detail would strengthen the paper.
Likewise, more statistical analysis that would support the cross-dataset claim is needed, beyond the simple MAE reported. For example, provide a deep analysis on the context of the 9.1 cm MAE. Is this error constant, or does it correlate with other factors (e.g., higher error for distant fixations, specific gaze angles, or movement patterns)?
Provide a figure that clearly visualizes the necessity of the proposed subject-invariant normalization. How does the distribution of the normalized features compare to the unnormalized features across different subjects?

---

### Official Review · Reviewer_v9vg · 2025-10-30

**Soundness:** 3
**Presentation:** 2
**Contribution:** 3
**Rating:** 4
**Confidence:** 3

**Summary:**

Paper proposed a novel LSTM method to estimate fixation depth from calibration-free gaze data. It employs a novel subject-invariant normalization strategy. Experimental results outperform SOTA method, mix-tcn and other baseline methods: GRU, 1D-CNN, TCN, and multi-head transformers for the Robust Vision dataset.

**Strengths:**

1. Originality. Paper has proposed a novel method for the depth estimation of fixation with the LSTM approach. The method is shown to be superior to the SOTA method and other baseline methods.

2. Quality. Paper is of high quality. It put up a clear hypothesis which is supported by the experimental results. The proposed method is clearly explained, and the extensive experiments were done to validate and investigate the proposed method.

3. Clarity. The paper's hypothesis, related works, methodology were clear. The experimental setup and design is also well described.

**Weaknesses:**

The presentation of experimental results are unconventional and confusing/incomplete. There was no table which directly compares the various baselines for each dataset. There were several references to Table 5.1, which is actually Table 1.

While it is shown in Table 4, the comparison between the different baselines. There were no other comparisons for the other datasets, Tufts, Gaze-In-the-Wild. It is unclear if the experiments were not done, or that the results were omitted.

The fixation depth estimation problem is rather niche and its significance is limited. This can be inferred from the lack of published works in this area.

**Questions:**

What's the comparison of the experimental results for proposed method and the baseline methods for the Tufts, Gaze-In-the-Wild, and cross-domain datasets?

The baseline methods used training data heavy transformer when there are more data efficient transformer methods, e.g. DeIT. Will the author also include comparison with such backbone?

**Details Of Ethics Concerns:**

No concern.

---

### Official Review · Reviewer_WB1s · 2025-10-30

**Soundness:** 2
**Presentation:** 1
**Contribution:** 2
**Rating:** 2
**Confidence:** 3

**Summary:**

The authors present a LSTM based method for estimating fixation depth during eye tracking. They present a featurization strategy to make this system robust to new participants. They compare MAE in a LOOCV task averaged across several datasets on a few baseline architectures, and one existing method in the literature. They analyze inter-subject variation and distribution of residuals in their model.

**Strengths:**

- Nice evaluation of results and investigation of error cases, inter subject variability, and residuals
Good baseline dataset selection in table 1
- Nice discussion of related work
- Nice statistical handling of ablation results, though I think the issue with these ablations are not the statistics but rather whether these models were properly tuned before reporting metrics
- The domain seems relevant to the research community

**Weaknesses:**

- This paper is a direct resubmission of a poorly performing NeurIPS paper. There are no changes in any of the writing. I reviewed it once and recieved no comments or modifications, my review and scores are the same as the first submission.

- Main dataset results (Table 1) do not compare to any baseline methods in the field
- Not much architecture innovation – it looks like an MLP on top of a max-pooled LSTM, if the preprocessing is the important part then it might be best to feature that more strongly and clearly in the work
- Architecture ablations in the supplement seem a bit suspicious – did they apply the same level of rigor in optimizing them via optuna that they did for their own method. In particular the transformer looks like its got a key bug stopping the fitting
- It seems the paper is padding a bit to get to the length, the main text of the work is a bit light on nontrivial contributions. Consider moving things like the impact statement and code release into the supplement and expanding on your featurization strategy which makes a better case for your novelty. You can also tighten up future contributions as it’s a bit long for a main section
- It would be better to use the whitespace for more important results rather than an analysis of errors which can go in supplement. Use this space to expand on your comparison of baselines
- Include a table with performance metrics across all of your datasets and baselines. Don’t just report numbers averaged over all datasets, include error bars on these numbers

**Questions:**

n/a

**Details Of Ethics Concerns:**

- This paper is a direct resubmission of a poorly performing NeurIPS paper. There are no changes in any of the writing. I reviewed it once and recieved no comments or modifications, my review and scores are the same.

https://openreview.net/forum?id=AtXs8X1kDA&referrer=%5BReviewers%20Console%5D(%2Fgroup%3Fid%3DNeurIPS.cc%2F2025%2FConference%2FReviewers%23assigned-submissions)

---

### Author Response · Authors · 2025-11-15
**Withdrawal of Submission**

We thank the reviewers for their constructive feedback. Due to unforeseen circumstances and the need for additional experiments, we won't be able to provide a full rebuttal in the given time. We appreciate the reviewers’ efforts and plan to Withdrawal of Submissionextend this work and resubmit it in the future substantially.

---

### Note · Authors · 2025-11-15

**Comment:**

We thank the reviewers for their constructive feedback. Due to unforeseen circumstances and the need for additional experiments, we won't be able to provide a full rebuttal in the given time. We appreciate the reviewers’ efforts and plan to extend this work and resubmit it in the future substantially.

**Withdrawal Confirmation:**

I have read and agree with the venue's withdrawal policy on behalf of myself and my co-authors.